# High-Speed Variable Polynomial Toeplitz Hash Algorithm Based on FPGA

**DOI:** 10.3390/e25040642

**Published:** 2023-04-11

**Authors:** Si-Cheng Huang, Shan Huang, Hua-Lei Yin, Qing-Li Ma, Ze-Jie Yin

**Affiliations:** 1National Synchrotron Radiation Laboratory, State Key Laboratory of Particle Detection and Electronics, University of Science and Technology of China, Hefei 230029, China; 2National Laboratory of Solid State Microstructures, School of Physics, Collaborative Innovation Center of Advanced Microstructures, Nanjing University, Nanjing 210093, China; 3College of Electronic Engineering, National University of Defense Technology, Hefei 230037, China; 4State Key Laboratory of Particle Detection and Electronics, University of Science and Technology of China, Hefei 230026, China; zjyin@ustc.edu.cn

**Keywords:** Secure Hash Algorithm, quantum digital authentication, variable irreducible polynomial, Field-Programmable Gate Array, Fast Modular Composition Algorithm

## Abstract

In the Quantum Key Distribution (QKD) network, authentication protocols play a critical role in safeguarding data interactions among users. To keep pace with the rapid advancement of QKD technology, authentication protocols must be capable of processing data at faster speeds. The Secure Hash Algorithm (SHA), which functions as a cryptographic hash function, is a key technology in digital authentication. Irreducible polynomials can serve as characteristic functions of the Linear Feedback Shift Register (LFSR) to rapidly generate pseudo-random sequences, which in turn form the foundation of the hash algorithm. Currently, the most prevalent approach to hardware implementation involves performing block computations and pipeline data processing of the Toeplitz matrix in the Field-Programmable Gate Array (FPGA) to reach a maximum computing rate of 1 Gbps. However, this approach employs a fixed irreducible polynomial as the characteristic polynomial of the LFSR, which results in computational inefficiency as the highest bit of the polynomial restricts the width of parallel processing. Moreover, an attacker could deduce the irreducible polynomials utilized by an algorithm based on the output results, creating a serious concealed security risk. This paper proposes a method to use FPGA to implement variational irreducible polynomials based on a hashing algorithm. Our method achieves an operational rate of 6.8 Gbps by computing equivalent polynomials and updating the Toeplitz matrix with pipeline operations in real-time, which accelerates the authentication protocol while also significantly enhancing its security. Moreover, the optimization of this algorithm can be extended to quantum randomness extraction, leading to a considerable increase in the generation rate of random numbers.

## 1. Introduction

The flourishing development of computer networks has greatly increased the amount of communication exchange, thereby resulting in various information security issues including information loss, leakage, and tampering. These issues not only pose a risk to personal privacy and corporate assets, but could also compromise national security and economic stability. As a primary measure to ensure information security, encryption and authentication technologies serve as critical barriers to protecting the confidentiality, integrity, and availability of information for individuals, enterprises, and even countries. Reliable and efficient information encryption systems are the goal of enterprises and the nation’s tireless pursuit, offering promising market prospects and significant value for economic and scientific research [1,2,3,4].

Secure hashing algorithms, also known as hashing algorithms, are characterized by their non-directional nature and strong collision resistance. They operate by mapping a given arbitrary-length keyword to a fixed-length hash value. These algorithms are commonly utilized in digital authentication, encryption, indexing, and quantum randomness extraction applications in QKD systems [5,6,7,8]. For digital authentication, it is necessary to compress an arbitrary-length input to a 128-bit hash value to facilitate the authentication process, and developing the rapid hash operations is critical to managing large data streams. Similarly, fast hashing operations are essential in quantum random extraction to extract the quantum randomness in the original data. These algorithms can effectively block illegal and spam messages and verify the integrity of data interactions, improving the overall security of the system [9].

For cryptography and encryption algorithms, the irreducible polynomial serves as the fundamental building block for constructing the hash algorithm, such as the widely used Secure Hash Algorithm (SHA). It functions as a characteristic function of the Linear Feedback Shift Register (LFSR) to generate pseudo-random sequences with high efficiency. It is worth noting that the hash algorithm cryptosystem has been demonstrated to be resistant to quantum attacks, as evidenced by [10,11,12].

Despite that the significance of irreducible polynomials and hash algorithms in the realm of information security is indisputable, the previous systems capable of quickly generating such entities can be further improved. Most of the current hash algorithm chips utilized in information security rely on a single algorithmic logic or a fixed irreducible polynomial. These approaches offer greater simplicity in constructing the algorithm and facilitate parallel processing. However, using a fixed irreducible polynomial as the characteristic polynomial of the LFSR can be vulnerable to being triggered by the irreducible polynomial coefficients when subjected to large data volumes. This vulnerability increases the risk of the algorithm being deciphered, presenting hidden dangers and failing to satisfy the growing need for enhanced information security.

To enhance both security and computational efficiency, it is necessary to employ variational irreducible polynomials for computations. Specifically, we can use the variable irreducible polynomials as the characteristic polynomials of LFSR. This approach can reduce the risk of decipherment through extensive computations, and the the algorithm’s security can be further improved through random shifts of polynomial factors. However, parallel processing of variable polynomials is challenging, and utilizing the previous algorithm as the characteristic polynomial of the LFSR considerably reduces operational speed, making parallel processing unfeasible. Therefore, it is crucial to develop variational and irreducible polynomial algorithms that can be optimized with hash algorithms for ensuring improved computational speed and enhanced security of the system.

## 2. Algorithm Optimization for FPGA Based Pipeline Generation of Toeplitz Matices

In the existing FPGA implementation scheme of the hashing algorithm, first, the FPGA transforms the data to be computed via FIFO. At each clock cycle of the FPGA, 64-bit data are read from the FIFO. At the same time, according to the set characteristic polynomial, a 64×128 Toeplitz matrix is generated to calculate the 64-bit data. In the case where the high 64-bit coefficients of the characteristic polynomial are all zero, all the computations can be performed in a single cycle, and the data can be processed in parallel in the form of a pipeline for real-time processing effects [13,14,15,16,17].

The algorithm flow chart is shown in Figure 1.

In this approach, pipelining can be performed in an FPGA, provided that the high-level 64-bit characteristic polynomial of the LFSR is all zero. Therefore, we have to fix an irreducible polynomial in the FPGA code as the characteristic polynomial of the LFSR. Doing so ensures that we can compute a 64×128 Toeplitz matrix in one clock period.

For large-scale data computation, we need to update the Toeplitz matrix in real time. One traditional method is to use an irreducible polynomial as the characteristic polynomial of the LFSR to generate pseudorandom numbers, which are then used to extend the Toeplitz matrix. However, as the amount of data increases, the pseudorandom numbers produced may exhibit periodic behavior, which can be exploited by attackers for malicious activities. This could pose significant security risks to the computation process. However, when the characteristic polynomial is random, if the high 64-bits of the characteristic polynomial are not zero, the traditional method will not be able to carry out parallel pipeline operation, which greatly reduces the computation speed.

Our method aims to address the security risks associated with using the traditional approach to update the Toeplitz matrix for large-scale data computation. To achieve this, we utilized the Fast Modular Composition Algorithm (FMC) algorithm to constantly generate new irreducible polynomials in FPGA. Then, we updated the characteristic polynomials of the LFSR in real-time during each hash operation.

One critical aspect of our approach is ensuring that the irreducible polynomials are quickly and efficiently computed in FPGA. By dynamically updating the LFSR polynomial, we can prevent the pseudo-random numbers from exhibiting periodicity and improve the overall security of the system. Another crucial area of focus is devising efficient strategies for performing hash computations when the characteristic polynomials of LFSR are changing continuously. This is necessary to ensure that the computation process remains fast and accurate despite the constant updates.

## 3. The Generated Algorithm of Irreducible Polynomials

### 3.1. The Characteristics of the Irreducible Polynomials

Irreducible polynomials are polynomials that cannot be written as a product of two polynomials of lower degree. That is, if p(x) is irreducible, then for any polynomial F(x),(p(x),F(x))=1. With irreducible polynomials as the characteristic polynomials of LFSR, the algorithm can generate uniform pseudorandom numbers that satisfy the application requirements of hash algorithms. However, irreducible polynomials cannot be computed directly. In order to be updated in real-time in the algorithm of irreducible polynomials coefficient, we have to judge the polynomial, decide whether it is an irreducible polynomial, whether it does not meet the irreducible polynomials judgment conditions to update the polynomial, until there is a way to generate the irreducible polynomials we need.

Thus, the speed at which irreducible polynomials are determined determines the superiority of our algorithm.

### 3.2. Conditions for Judging Irreducible Polynomials

The GF(2) represents a finite field with two elements, denoted by 0 and 1, respectively. In the GF(2) domain, addition follows the operation rules of XOR.

Equation (Equation 1) shows the necessary and sufficient conditions for the irreducibility of a polynomial P(x) of order N in the codomain of GF(2) [18]:(1)$$\left\{\begin{array}{c} \left(1\right)x^{2n}=x\;\mathrm{mod}\;p\left(x\right)\\ \left(2\right)\mathrm{GCD}\left(x^{2^{\frac{m}{d}}}-x,p\left(x\right)\right)=1,disanypimefactorofN \end{array}\right.$$
Condition (1) guarantees that p(x) is the product of irreducible polynomials of order *n* or less than order *n*; condition (2) requires that the greatest common factor of p(x) and $x^{2^{\frac{m}{d}}}-x$ be 1, which rules out that p(x) has irreducible polynomials of an order less than *n* as factors.

Verification condition (2) can directly find the common factors of $x^{2^{\frac{m}{d}}}-x$ and p(x). When *N* is a power of a prime number, only one calculation of the largest common factor is needed. For example, $n=128=2^{7}$ has only a prime factor of 2, and only the common factors of $x^{2^{64}}-x$ and p(x) need to be calculated.

However, for higher order terms in *x*, the computation of the modulus will be extremely complicated and difficult to implement fast for FPGA. In order to reduce the amount of calculation and increase the speed of calculation, we used the Fast Modular Composition (FMC) algorithm, through a tiny number of iterations that can be solved for the value of $x^{2^{n}}\;\mathrm{mod}\;p\left(x\right)$.

### 3.3. Fast Modular Composition Algorithm

The following is a quick way to calculate $x^{2^{n}}\;\mathrm{mod}\;p\left(x\right)$ [19]:

First, calculate $x^{2}=x^{2^{2^{0}}}\;\mathrm{mod}\;p\left(x\right)$, then use the Fast Modular Composition (FMC) algorithm for ${\left(x^{2}\right)}^{2}=x^{2^{2}}=x^{2^{2^{1}}}\;\mathrm{mod}\;p\left(x\right)$; further use the FMC algorithm to calculate ${\left(x^{2^{2}}\right)}^{2^{2}}=x^{2^{2^{2}}}\;\mathrm{mod}\;p\left(x\right)$. For $n=128=2^{7}$, you can start from $x^{2}$ and repeat FMC seven times to get $x^{2^{128}}=x^{2^{2^{7}}}\;\mathrm{mod}\;p\left(x\right)$. Similarly, for the $2^{n}$-bit hash algorithm, we can also quickly calculate its corresponding irreducible polynomial by using the FMC algorithm through n iterations. Since the 128-bit hash algorithm already has a sufficient level of security, this article only discusses the processing flow at 128 bits.

FMC algorithm: input p(x), s(x)modp(x), output s(s(x))modp(x):Let $m=\lceil \sqrt{n}\rceil$ (rounded up with respect to $\sqrt{n}$), and let $s\left(x\right)=\sum_{i=1}^{m-1}s_{i}\left(x\right)x^{mi}$ (where $s_{i}\left(x\right)x^{mi}$ is actually the degree of s(x) in the interval $\left[m_{i},m_{(}i+1)-1\right]$);For 2≤i≤m, we compute $s{\left(x\right)}^{i}\;\mathrm{mod}\;p\left(x\right)$;Define an m×n matrix *A* on $F_{2}$ whose rows are the coefficients of $1,s\left(x\right)\;\mathrm{mod}\;p\left(x\right),\ldots ,s{\left(x\right)}^{m-1}\;\mathrm{mod}\;p\left(x\right)$. Define the m×m matrix *B* on $F_{2}$, and the i+1(0≤i≤m−1) row of *B* is the inverse of the coefficients of $s_{i}\left(x\right)$. Calculate C=BA;For 0≤i≤m−1, $C_{i}\left(x\right)$ denotes the polynomials with the i−th row of *C* as coefficients and computes $b=\sum_{i=0}^{m-1}C_{i}\left(x\right)s{\left(x\right)}^{mi}\;\mathrm{mod}\;p\left(x\right)$;Output *B*, *B* is s(s(x))modp(x).

In this algorithm, input $x^{2}$ and run it to get the result, then input the polynomials obtained from the last run repeatedly to get $x^{2^{2^{1}}}\;\mathrm{mod}\;p\left(x\right),x^{2^{2^{2}}}\;\mathrm{mod}\;p\left(x\right),\ldots ,x^{2^{2^{n}}}\;\mathrm{mod}\;p\left(x\right)$ in turn, and $n=2^{\log_{2}n}$. After $\log_{2}n$ iterations of the FMC algorithm, we will get $x^{2n}=x^{2^{2^{\log_{2}n}}}$. If *n* is a power of two, notice that $\frac{n}{2}=2^{\log_{2}n-1}$, running the FMC algorithm for the $\log_{2}n-1$ time will give you $x^{2^{\frac{n}{2}}}\;\mathrm{mod}\;p\left(x\right)=x^{2^{2^{\log_{2}n-1}}\;\mathrm{mod}\;p\left(x\right)}$. So there is no need to perform any more calculations.

### 3.4. Generate Irreducible Polynomial in FPGA Based on Fast Modular Composition Algorithm

According to the FMC algorithm described above, the logic for generating irreducible polynomials is divided into several parts.

Stochastic generation of polynomials: One can use a preset seed to randomly generate a polynomial in the FPGA and import its coefficients into the next judgment module;Determine the coefficients of the imported polynomial. According to the FMC algorithm, the imported coefficients are passed through the square operation module, the modulus computation module, the counting judgment module, the cyclic modulus computation module, and the result judgment module. The square operation module performs square operations on the input data, that is, calculates the value of $x^{2^{n}}$; the modulo computation module performs a modulo computation on the input square operation module data and the input irreducible polynomial parameters and computes $x^{2^{n}}\;\mathrm{mod}\;p\left(x\right)$; the counting judgment module is used to record the times of the square operation module and the modulus computation operation module, and the data results for specific times are recorded based on the judgment results. In this project, a 128-bit hashing algorithm was used, so the number of cycles here is n=7; the cyclic modulo computation module performs a cyclic modulo computation on the input count judgment module data and the input irreducible polynomial parameters; the module of the resulting judgment determines whether the irreducible polynomial parameters satisfy the requirements. If the irreducible polynomial argument does not satisfy the requirement, the operation of adding the number 2 to the irreducible polynomial argument is performed, and the irreducible polynomial identity operation is repeated until the irreducible polynomial argument satisfies the requirement.

The specific algorithm flow chart is shown in Figure 2.

## 4. Implementation Flow of Hashing Algorithm Based on Variable Characteristic Polynomial in FPGA

To improve the security, an irreducible polynomial will be randomly generated as the characteristic polynomial in this study. If there is one entry in the higher 64 bits of the generated characteristic polynomial, the Toeplitz matrix will not be able to complete all the operations in a single cycle, which will considerably slow down the hash operation [20,21,22,23].

To address the above issues, the optimization of the hashing algorithm was carried out in this study: before the pipeline started, the equivalent polynomials were computed and derived from the randomly-generated characteristic polynomials. The effect is equivalent to a process of forcing the expansion of the original LFSR formula in the presence of the nonzero coefficients of the characteristic polynomial with 64 bits or more. The so-called forced unfolding procedure means that, when the LFSR is used for computation, the LFSR of the previous level is iterated to this level for computation if the result of the previous level’s computation is needed. This is shown in the following Equation (Equation 2):(2)$$\left\{\begin{array}{cc} p\left[i+1\right]=p{\left[0\right]}^{\wedge }\left(p\left[i\right]\ll 1\right), & p[i][127]=1\\ p[i+1]=p[i]\ll 1, & p[i][127]=0. \end{array}\right.$$
After computing 64 equivalent polynomials, real-time calculations can be performed following the previous pipeline procedure. The treatment flow diagram of the system and the diagram of the derivation procedure for equivalent polynomials are shown in Figure 3 and Figure 4.

Where “P_0” is the input irreducible polynomial, “P[0] - P[63]” is the calculated equivalent polynomial, and “i” is the cyclic count. Through the judgment of the highest bit (P[i][127]) of the polynomial, the corresponding 64 equivalent polynomials are calculated iteratively.

In this approach, irreducible polynomials are forced to be expanded in advance by means of shift and XOR methods. Each newly generated irreducible polynomial needs only one expansion to compute the corresponding 64 equivalent polynomials. Toeplitz matrices can be generated in the FPGA pipeline in real time to process the input data:First 64 clock cycles: forcibly expand the input polynomials and calculate their 64 equivalent polynomials;Sixty-fifth clock cycle: read the first data to be processed, D1;Sixty-sixth clock cycle: read the second pending data D2 and compute the hash value T1 corresponding to the first 64-bit data;Sixty-seventh clock cycle: reads the third data D3 to be processed, computes the hash value T2 corresponding to the second 64-bit data, and accumulates the first hash value T1 to the output register *H*;Sixty-eighth clock cycle: read the fourth data D4 to be processed, compute the hash value T3 corresponding to the third 64-bit data, and accumulate the second hash value T2 to the output register *H*;After all the data to be processed have been processed, the final hash value, *H*, is output.

Since the equivalent polynomial is computed before the pipeline starts, the real-time performance of the algorithm is unaffected.

Moreover, to further improve the speed, 128 equivalent polynomials can be pre-computed to further improve the bit-width for parallel processing in the original method, which can also satisfy real-time hash operations under any characteristic polynomial.

## 5. FPGA Algorithm Implementation Verification

Aiming at the algorithm we proposed, we developed the corresponding hardware system to implement the algorithm. The algorithm implementation validation system block diagram is shown in Figure 5.

In the design, the power supply system supplies power to each module of the board. The FPGA clock is provided by an external chip. FPGA is responsible for algorithm implementation and data communication. Data input and output can be implemented through the PCIE bus or the SFP interface. According to the requirements of different I/O rates, the data to be hashed can be input to the FPGA through the PCIE bus or SFP interface. The FPGA hashes the data and outputs the calculation result.

In the FPGA, we used the FMC algorithm to generate irreducible polynomials and hash the generated irreducible polynomials as the characteristic polynomials of LFSR. The program framework in FPGA is shown in Figure 6.

As shown in the figure, the FPGA first obtains a random number seed through the communication module, which is used as the initial condition for generating irreducible polynomials. After the random number seeds are fed into the FMC algorithm module, the module computes irreducible polynomials satisfying the requirements as the characteristic polynomials of the subsequent LFSR. The characteristic polynomial coefficients are then fed into the equivalent polynomial computation module to compute the corresponding equivalent polynomials. Finally, these equivalent polynomials are used for pipeline computation in the FPGA, and the computed results are output through the communication module.

In our design, there are many matrix operations and multiplication on the GF(2) domain. When we carried out the RTL implementation, in order to meet the need of fast parallel operation, we used many LUT resources in FPGA to carry out the XOR operation. By splitting the matrix operations into parallel multiplication and addition operations, we can greatly improve the speed of calculation.

After completing the design of the FPGA program, we used Vivado software to evaluate the internal resource consumption of FPGA. The evaluation results are shown in Figure 7.

As can be seen, our chosen FPGA model can well meet our hardware resource requirements. We set aside more resource margins so that the Vivado software could easily complete the task of layout and wiring to meet the timing requirements.

At the same time, we also used Vivado software to analyze the power consumption. The program we designed corresponds to a total on-chip power of 6.711w, a junction temperature of 36.9 °C, and a thermal margin of 48.1 °C.

## 6. FPGA Implementation Performance Test

### 6.1. FPGA Calculation Rate Simulation Test

The simulation program was developed using the verilog language, which is suitable for FPGA, and Vivado2019.2 was used for compilation and simulation verification. Finally, the tests were performed on a K7 series FPGA chip from Xilinx, xc7k325tffg900.

First, the FPGA logic was simulated and tested through the simulation capabilities of the Vivado software. The simulation results are shown in Figure 8.

In the figure, “data” is the input data to be authenticated, “input_polynomial” is the variable polynomial of the input, “tag” is the hash value of the final output. Other signals are control signals for the computational process.

As can be seen in the figure, the simulation results indicate that the FPGA can process 128 bits of input data in real-time in each clock cycle when the clock input period is 100 MHz. In other words, this algorithm can compute a 128-bit hash value in 10 ns. The hardware computation speed of the algorithm can be calculated as 12.8 Gbps.

The logic is then written into the FPGA and connected to the host computer via the PCIE bus for rate testing.

Manually add a clock count to the logic, input a 128 MB file for the board to execute the hash operation, start counting when the PCIE detects the command, and stop counting when the FPGA completes the computation. Counting the total number of clock cycles required for this operation yields the time required to process the 128 MB file. The measured pure computation speed on the FPGA side is 10.88 Gbps.

### 6.2. PCIE Reading and Writing Test and Optimization

The actual pure computation speed on the FPGA side is 10.88 Gbps. Combined with the input data write DDR time and the processing time of the outage response, and the PCIE standard read and write speed of 22.4 Gbps, the theoretical speed of the entire hash computation can be calculated to be about 7.2 Gbps after subtracting the time spent writing DDR and processing the outage.

Connect the board card to the top computer, given a 134 MB file for the board to process, from the host computer to send instructions to the board to start timing, statistics from the host computer in the board to obtain the uploaded calculation results of the time needed, and then test the processing speed of the board.

Optimizing the read/write logic of BRAM_A and the control logic of xdma. It has been tested 8000 times and the results show that the operation is stable with a speed of about 6.8 Gbps, which is consistent with the predicted value. The errors mainly come from the instability of the PCIE read/write speed and the outage response speed.

By comparing the test results with reference [13,18], the existing methods only use fixed irreducible polynomials as characteristic polynomials, and the calculation speed reaches 1 Gbps. It can be seen that our algorithm greatly improves the speed of hash operations while optimizing a fixed irreducible polynomial in the reference variable. Using the same test data computed by the software, our algorithm is also ten times faster than the software algorithm.

## 7. Conclusions

In this paper, we present an improved method for generating irreducible polynomials and optimizing existing hashing algorithms through the use of variational polynomials. Our approach utilizes field-programmable gate arrays (FPGAs) and proposes the Fast Modular Composition algorithm to quickly compute the modulus of high degree polynomials.

We optimized the original hashing algorithm and proposed a parallel pipeline algorithm by computing the characteristic polynomial. The FPGA’s large resources and parallel operation increase the speed of hash operations and enhance system security by using variable irreducible polynomials as the characteristic polynomials of the hash algorithm.

Our proposed algorithm achieves a hashing speed of 6.8 Gbps based on variational irreducible polynomials on a single board while improving security compared to the traditional approach that uses fixed irreducible polynomials. This not only improves the speed of operation but also enhances security, resulting in a better performance of authentication protocols in Quantum Key Distribution (QKD) networks.

The hashing algorithm used in this paper has a length of 128 bits, but it can be modified to meet the requirements of other application scenarios by changing its length. However, it is essential to ensure that the number of bits m satisfies the condition that m is equal to 2 raised to the power of n, since failure to meet this condition will result in a reduction of the calculation speed of the Fast Modular Composition algorithm. It should be noted that a clock frequency of 100 MHz was utilized in this study, but if higher processing requirements are needed, the clock frequency can be increased to achieve faster hashing operation rates.

Furthermore, when hashing algorithms with a greater number of digits are utilized, the width of the data that is processed in parallel can be increased to further enhance operation speed.

## Figures and Tables

**Figure 1 entropy-25-00642-f001:**
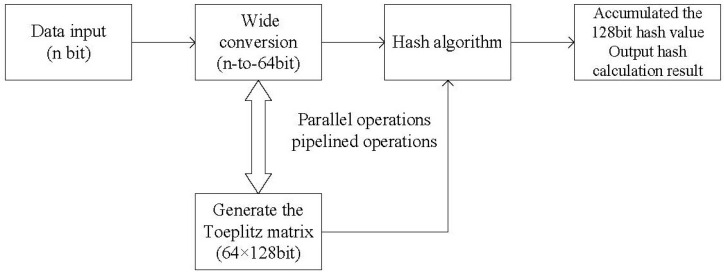
Algorithm flow chart of the original hash algorithm.

**Figure 2 entropy-25-00642-f002:**
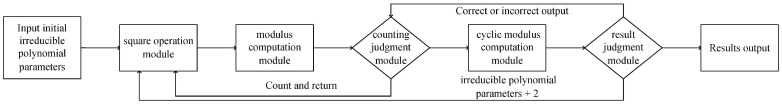
Algorithm flow chart of FMC.

**Figure 3 entropy-25-00642-f003:**
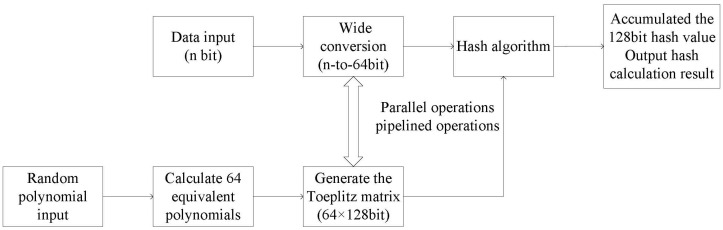
Flow chart of the optimized hash algorithm.

**Figure 4 entropy-25-00642-f004:**
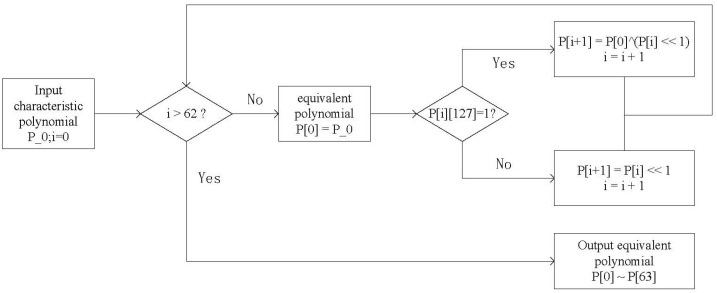
FPGA simulation test result.

**Figure 5 entropy-25-00642-f005:**
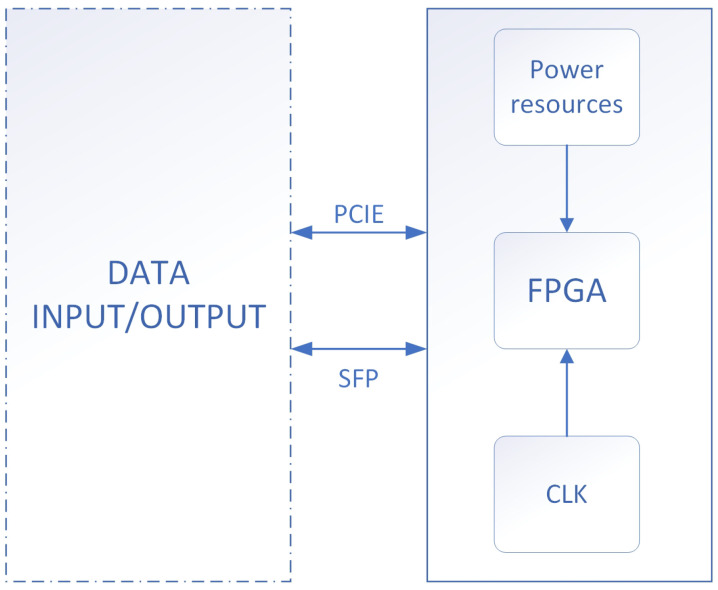
The algorithm implementation validation system block diagram.

**Figure 6 entropy-25-00642-f006:**
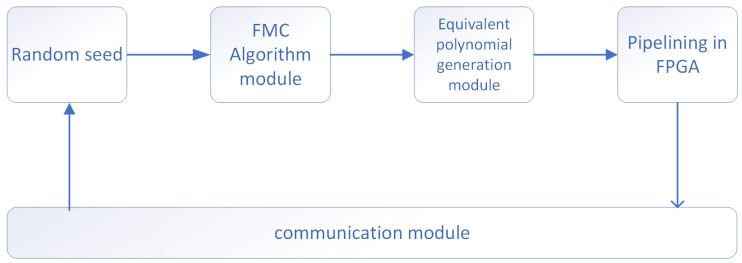
The program framework in FPGA.

**Figure 7 entropy-25-00642-f007:**
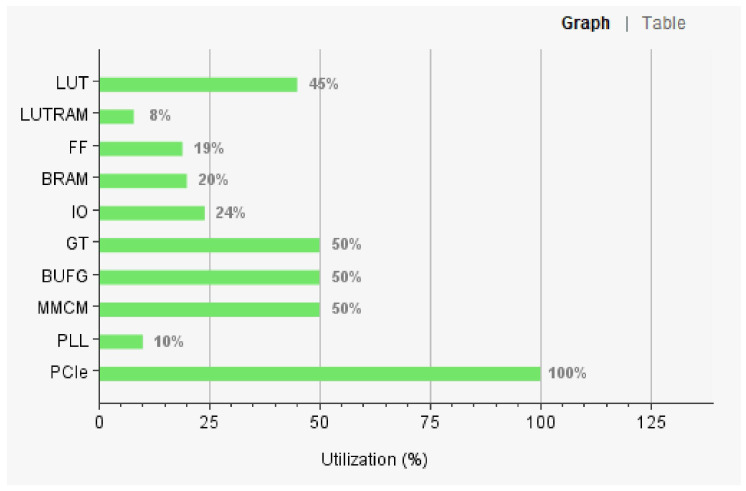
The internal resource consumption of FPGA.

**Figure 8 entropy-25-00642-f008:**
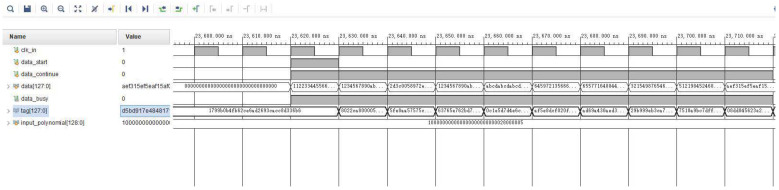
Algorithm flow chart of FMC.

## Data Availability

The data presented in this study are available on request from the corresponding author. The data are not publicly available due to [raw data needing to be de-encrypted].

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
