# Peer review of "High-Speed Variable Polynomial Toeplitz Hash Algorithm Based on FPGA"

_entropy, 2023, doi:10.3390/e25040642_

Round 1
Reviewer 1 Report
The focus of this manuscript was to provide an optimisation of the hash algorithm based on FPGA. More specifically, the authors propose to use equivalent polynomials based on randomly generated characteristic polynomials before the pipeline starts. So that the processing of the hashing algorithm (for authentication) can be accelarated.
I think major revision of this manuscript is needed before assessment for its publication in Entropy. Some comments are listed below:
1. The structure of the manuscript seems not properly prepared.
The authors spend quite some time explaining the irreducible polynomials and fast modular composition algorithm. Are they the equivalent polynomials mentioned later on? It is not clear when reading the manuscript. Most of the manuscript is describing background, while the details of this work only takes a small part.
2. More details of the proposed method and implementation are needed.
3. In the performance test section, how does the proposed method comparing to conventional methods?
4. In introduction section, the authors mentioned that Hashing needs to satisfy security requirement. How does the proposed method comply the security requirment? I would suggest some discussion on this.
5. There are quite a few typos and grammar mistakes in the manuscript. English should be improved for a better readability.
Reviewer 2 Report
QKD plays an important role in secure communication.
In this manuscript, the authors used the FPGA to implement variational irreducible polynomials based on the hashing algorithm. By computing equivalent polynomials and updating the Toeplitz matrix with pipeline operations in real time, our method accelerates the speed of authentication protocols to an operation rate of 6.8 Gbps, and considerably enhances the security of the algorithm. This work is important and it will have potential application in quantum key distribution. This work can be published in Entropy after some minor revisions. Some related work in QKD are suggested. For example,
[1]Bin Liu, Shuang Xia, Di Xiao, Wei Huang, Bingjie Xu, and Yang Li,
Decoy-state method for quantum-key-distribution-based quantum private query, Sci. China Phys. Mech. Astron. 65, 240312 (2022)
[2]W. Cui, et al., Satellite-based phase-matching quantum key distribution, Quant. Inf. Process. 21, 313 (2022)
[3]L. W. Hu, et al., Practical measurement-device-independent quantum key distribution with advantage distillation, Quant. Inf. Process. 22, 77 (2023)
[4]Q. Q. Peng, et al., Satellite-to-submarine quantum communication based on measurement-device-independent continuous-variable quantum key distribution, Quant. Inf. Process. 21, 61 (2022)
[5]L. C. Lwek, et al., Chip-based quantum key distribution, AABBS Bulletin, 31, 15 (2021).
Reviewer 3 Report
The title of the paper is "High-speed Variable Polynomial Toeplitz-hash Algorithm Based on FPGA" however in the paper, there is no track of hardware architectures, RTL systems, and considerations regarding power resources and circuits. It seems more software oriented than hardware oriented. In addition the comparisons with the state of the art are not clear
Round 2
Reviewer 1 Report
Based on the authors replies, there are two main contributions in this work. The first is the implementation of the Fast Modular Composition(FMC) algorithm in FPGA. I assume this is because typical implementation of such hash functions in FPGA is not as rigorous as required by its security assumptions? While the second one is to use the irreducible polynomials we obtain as characteristics polynomials in subsequent hash algorithm for the hash operation.
However, these motivations seem not clearly reflected in the revised manuscript, and I feel it's difficult to catch the flow and evaluate the significance of this work.
Besides, there are also some wrong conclusions and typos in the manuscript. For example, in line 34, there is one weak collision resistance and one strong collision resistance when describing the features of hashing algorithms. In line 41, the hashing in quantum randomness extraction is not to enhance the rate of obtaining random numbers.
I would suggest the author to further polish the manuscript, probably re-arrange its structure, state explicitly what is the problem of conventional methods to make your motivations clear to readers.
Reviewer 3 Report
Although the title of the paper suggests a work-oriented on FPGA implementation, the paper does not deal with the typical aspect of FPGA implementation as STA, hardware resource utilization, and power consumption. In addition, no information about the RTL implementation is provided.
Round 3
Reviewer 1 Report
I have no more comments.